# Happy Birthday? Relative Age Benefits and Decrements on the Rocky Road

**DOI:** 10.3390/sports10060082

**Published:** 2022-05-24

**Authors:** Neil McCarthy, Jamie Taylor, Andrew Cruickshank, Dave Collins

**Affiliations:** 1Premiership Rugby Ltd., 7th Floor, Regal House, Twickenham TW1 3QS, UK; 2Grey Matters Performance Ltd., Stratford upon Avon CV37 9TQ, UK; jamie@greymattersuk.com (J.T.); andrew@greymattersuk.com (A.C.); dave@greymattersuk.com (D.C.); 3School of Health and Human Performance, Faculty of Science and Health, Dublin City University, D09 Dublin, Ireland; 4Holyrood Campus, Moray House School of Education and Sport, The University of Edinburgh, Edinburgh EH8 8AQ, UK

**Keywords:** talent identification, talent development, challenge

## Abstract

(1) Background: There is abundant literature in talent development investigating the relative age effect in talent systems. There is also growing recognition of the reversal of relative age advantage, a phenomenon that sees significantly higher numbers of earlier born players leaving talent systems before the elite level. However, there has been little investigation of the mechanisms that underpin relative age, or advantage reversal. This paper aimed to investigate (a) the lived experience of relative age in talent development (TD) systems, (b) compare the experience of early and late born players, and (c) explore mechanisms influencing individual experiences. (2) Methods: interviews were conducted with a cohort of near elite and elite rugby union players. Data were subsequently analysed using reflexive thematic analysis and findings considered in light of eventual career status. (3) Results: challenge was an ever-present feature of all players journeys, especially at the point of transition to senior rugby. Psycho-behavioural factors seemed to be a primary mediator of the response to challenge. (4) Conclusions: a rethink of approach to the relative age effect is warranted, whilst further investigations of mechanisms are necessary. Relative age appears to be a population-level effect, driven by challenge dynamics.

## 1. Introduction

Effective and efficient talent identification and talent development (TD) processes are a significant part of the strategic management of TD systems. Increasing curiosity and investigation of such elements is a significant challenge for many national governing bodies (NGBs). TD systems are under increasing scrutiny, with data challenging established paradigms in relation to many TD dynamics [1]. Of significant debate are the dynamics pertaining to selection and development of athletes as they journey into, through and out of talent systems [2,3]. Whilst the accurate prediction of future performance has been a topic of significant research, practical application of this is significantly challenged by the biopsychosocial complexities of development [4,5]. This is especially so in the earlier years of talent development, with a variety of dynamics apparent, especially at selection gateways [6].

One such factor suggested as underpinning these selection biases is the relative age effect (RAE) [7,8]. The inevitable chronological grouping of children as they enter the education system has been shown to promote early advantage for those born just before or after the academic cut-off date (11). This mechanism for selecting children continues as they enter organised sport and talent systems. An abundance of literature highlighting asymmetric birthdates during selection processes has linked RAEs to maturation and the comparative advantages and/or disadvantages of being born one side or the other of the selection cutoff date within sport (typically Sept 1st in the United Kingdom, for a review see [9]). To date, much of the literature has focused on the disproportionate volume of players born in the first two quartiles (Q1 and Q2) of the selection year in comparison to those born towards the end of the selection year (Q3 and Q4).

Explanations for these effects have tended to focus on advanced physical maturation offering relatively older individuals up to 12 months advantage over their relatively younger peers [8,10]. Less reported, however, are the advantages/disadvantages that have been identified in other domains such as the cognitive and emotional disruptions observed during formative developmental periods [11,12,13,14,15]. Perhaps most importantly, we know little about how these biopsychosocial dimensions manifest in TD systems, especially given that RAE and maturation are acknowledged as separate constructs [16].

The orientation for the majority of RAE literature in sport has led to a focus on the potential negative effects age groupings have on the identification of individuals and their experiences [17]. These studies have generally focused on a specific moment in time for data collection (e.g., selection into talent systems) and thus offer limited perspective on long term effects. The general consensus is an assumption that the RAE is something to eradicate, to prevent large numbers of performers being excluded [18].

### 1.1. RAE Advantage Reversal

More recently, the literature has challenged these assumptions and has begun to report a potential positive that emerges from the attritional and/or challenging experiences of the relatively young. This has separately been identified as the ‘underdog hypothesis’ [19] and ‘advantage reversals’ [1,20], identifying that whilst a disproportionately high number of early birthday athletes are initially selected, the relatively young are proportionately more likely to reach senior elite status. This finding appears robust across a wide range of sporting contexts: in handball [21], cricket [22], ice hockey [23] and across male elite sport [24]. Indeed, highlighting the robustness of the finding, replications have consistently shown the same finding. For example, recent findings [8] show evidence of the same RAE advantage reversal previously found in a single academy [20] and across international pathways in rugby union and cricket [1]. Therefore, it appears that whilst those with early advantages are being selected into the initial stages of talent systems in greater proportions, earlier born athletes are leaving in far higher numbers than later born. Importantly, this ‘advantage reversal’ does not suggest a reversal of the RAE. Instead, it shows that those born later in the selection year are less likely to be deselected than their earlier born counterparts.

### 1.2. Mechanisms

This would suggest that the current literature base is limited by some key assumptions and by a lack of mechanistic focus. Those who are born earlier in the selection year are more likely to be selected for a TD system. It also appears that, at the population level, their relatively younger peers are more likely to continue through the TD system to elite performance. Yet, to this point, much of the extant literature has focused on ‘solving’ a variety of early advantage effects by focusing on levelling the playing field, for example: bio-banding [25]; age order shirt banding [26]; birthday banding [27], performance banding [28] and corrective adjustment procedures [29]. Yet, very little attention has been paid to the dropout rates of those with earlier advantage [30] and investigation of underpinning mechanisms is disappointingly sparse. As a result, we know that the RAE exists and that there is likely to be a reversal of advantage, but we do not know why this happens. This is a key barrier for the practitioner seeking to optimise TD processes. McCarthy and Collins [1] suggested a potential mechanism could be the initial impact of negative selection experiences, with these early disadvantages being facilitative of greater psychological ‘growth’ and/or acting as a mechanism for a more intrinsically focused and longer-term motivational orientation This hypothesis suggested that RAE advantage reversals may be driven by the motivational orientation of individuals and how that is anchored through formative experiences.

Motivation is a significant factor in sports participation, progression and drop out [31,32,33]. One underpinning feature of an individual’s motivation is perceived competence. Perceived competence acts as a domain-specific indicator of self-esteem that contributes to and is affected by the individual’s motivational orientation [34,35]. For example, the relatively old may progress rapidly as a result of early challenge-free experiences, arriving at early selection gateways with a high degree of perceived competence. This may inadvertently develop an individual with extrinsically anchored motivational orientation. Conversely, those athletes not afforded this advantage may develop a more intrinsic motivational orientation, becoming more likely to remain and persevere within TD systems. As such, later drop out from TD systems is proportionately higher from relatively older cohorts [8], suggesting these individuals may not be sufficiently orientated and/or equipped to cope with and prosper motivationally through transitions when early advantages begin to disappear [20,36]. Notably, this hypothesis seems to marry with other research suggesting a complex interaction between challenge and psycho-behavioural skills [2] and the risk posed to progression when there is a mismatch between the two [33].

Accordingly, there appears to be increasing evidence pointing to the interaction of challenge and psycho-behavioural skills [37,38,39]. This need for athletes to be challenged in their development has been accepted from a variety of research perspectives [30,40,41]. Key differences between these positions notwithstanding, it appears that performers who bring a variety of psycho-behavioral resources to challenging periods will be more likely to cope with and learn from their experience [42]. In this regard, recent investigations have suggested that challenge-filled sporting pathways are an essential feature of developmental journeys [43]. Further, it appears that sporting ‘traumas’ and/or challenging experiences, rather than being directly causative of ‘psychological growth’, instead act to test, prove and encourage previously developed psychological skills [42]. Indeed, perceptions of control, confidence and perspective, underpinned by psycho-behavioural skills [38], along with appropriate reflection and social support, appear particularly important in this regard [37,44]. This also appears to be the case amongst the limited populations where the reversal of relative advantage has been tested [45].

Reflecting these complexities, and beyond establishing RAEs and consequently advantage reversals in different contexts, there is a need to understand the mechanisms at play [23]. This is especially the case for the applied practitioner or policy maker who needs to make decisions regarding sporting systems, placing the individual performer’s experience as a primary concern [43]. Accordingly, this paper aimed to (a) generate a deeper understanding of the lived experience of the RAE in TD systems across groups of more and less successful athletes, (b) compare the experience of early and late born players, and (c) explore the mechanisms influencing individual experiences.

## 2. Materials and Methods

### 2.1. Research Philosophy

Grounded in the real-world factors covered in our introduction, alongside our desire to deepen knowledge in RAE for practical purposes, a pragmatic philosophy was adopted for the present study [46]. The primary objective of pragmatic research is to generate knowledge that is practically useful for the individuals and groups that it studies, plus the practitioners who support them [47]. Ontologically, pragmatism therefore requires researchers to avoid seeking universal truths or entirely subjective constructions and to instead identify processes and mechanisms that shape common experiences in specific settings at specific times [46]. Epistemologically, pragmatism is also based on the idea that a continuum exists between more objective and more subjective perspectives. Rather than posing questions against a pre-set epistemology, pragmatists therefore place their questions at the heart of a study and select an epistemological position and methods that are appropriate to answering it [48].

Based on the aims established in our introduction, including the need to move understanding of RAE beyond statistical phenomena, an interpretivist epistemology and qualitative strategy were selected for the present study [49]. More specifically, these approaches reflected our intention to understand experiences of the RAE, from the views of a relevant—and internally diverse—group of individuals [50,51]. Importantly, pragmatism also views researchers as part of the world they explore and encourages them to actively interact with the experiences of their participants in the knowledge generation process [46,52,53]. In this respect, all parts of this study were aided by the research team’s record of performing and working in elite sport and—with direct relevance to the participants in this study—elite rugby union specifically. Most notably, the first author was involved in elite rugby union as a player, then coach and TD practitioner; the second author as a coach, TD practitioner and coach developer; and the third and fourth authors as psychologists and coach developers.

### 2.2. Participants

To explore experiences of the RAE, eight male players who had entered the academy system in English Premiership rugby union and reached the transition point to the professional game (i.e., the final academy phase) were purposefully sampled via the contacts of the first author. At the time, he was an Academy Director at a professional club. To avoid the risks of collecting data overly influenced by situational factors, participants were also selected on the basis that they had gone through the academy system across various periods of time (rather than all coming from one cohort, or close cohorts). As such, data were collected from participants who had transitioned out of the academy programme at different times over the course of eight seasons. For sufficient comparison of experience (as per our third aim) individuals were also identified on the basis that their birthday was at either end of the sport’s selection year (i.e., Q1: between September and November; Q4: between June and August).

At the time of interview, all players were aged between 21–32 years of age (M = 26.5, *N* = 8) and actively involved as professional players either at an English Premiership (if retained) or Championship team (if released after reaching the end of the academy phase). Details relating to each specific participant’s birth quartile, initial, mid and overall career status (the latter using criteria from two) are provided in Table 1. Initial career status was determined by contract status when leaving the academy programme. Mid-career status was determined by analyzing each participant 5 years post transition from the academy. In all instances: ‘Championship’ refers to the second division of English rugby, ‘Premiership’ is the highest level of the domestic game in England and ‘senior test’ is a player who has played at international level. Please note additional information is limited to protect anonymity.

### 2.3. Data Collection

Prior to data collection, ethical approval was obtained from the first author’s institutional ethics committee and informed consent gained from each participant. All interviews were conducted by the first author who began by asking participants to plot their career trajectory on a gridded timeline. More specifically, the *X*-axis spanned the participant’s first involvement in sport all the way to the date of interview; and the *Y*-axis represented the participant’s perceived level of development and performance throughout this time [54,55]. Participants were then asked to highlight particularly critical periods and events along this timeline [37]. Using these timelines to minimize the limitations of retrospective recall [56], particularly for those who had moved from an academy to full professional contract a number of years previously, interviews were constructed against a semi-structured guide focused on key transitional periods and individual experiences for each participant. Consisting of open-ended questions and follow-up probes and prompts that were informed by RAE and TD literature, the guide was designed to offer flexibility for participants to describe their experiences in bespoke ways, while ultimately remaining focused on the study’s aims and principles reported in prior research [57]. Examples of main interview questions were: ‘Looking at your timeline can we discuss the points in your journey that were challenging and the points where you were finding it easy?’ and ‘What are your reflections on your progression now you have transitioned through the academy?’. All interviews took place face to face, lasting between 45 and 90 min (M = 66.4), and were audio recorded.

### 2.4. Data Analysis

Following data collection, all interviews were transcribed verbatim and analyzed using QSR NVivo software. Coherent with our desire to understand lived RAE experiences, plus the meanings attributed to this by participants, a reflexive thematic analysis (TA) was chosen as the specific analytic strategy [50]. Similarly, TA was also coherent with our pragmatic philosophy in that this form of analysis recognizes that researchers are a resource to support the interpretive process [58]. For the purpose of comparison between birth quartiles, players were grouped based on selection and school year. For participants from the first birth quartile (*N* = 4) and the fourth quartile (*N* = 4), players were further grouped according to initial career status (*N* = 2, retained and *N* = 2, released) from each birth quartile.

Based on the established TA process [59], analysis was undertaken in a recursive and blended fashion (based on the experience of the research team: [58]) and began with the first author reading through each transcript to optimize familiarity with the data and note early points of interest. This was followed by the application of codes to meaningful sections of raw data. More specifically, these codes were either semantic (to capture surface meaning) or latent (utilizing pre-exiting theories to interpret meaning: [50]). The third step saw the generation of initial themes, with significant codes being promoted to a theme, or similar codes being clustered together as patterns of shared meaning [50]. The fourth step involved a review of initial themes, after which the fifth and final step was taken to generate overarching themes and the final thematic map [60]. Importantly, whilst the first five phases of analysis were completed in the period following the collection of data, the sixth and final stage of TA, the write up, was delayed until each participant was 30 years old to take account of eventual career status of the participants.

### 2.5. Trustworthiness

As well as the approaches detailed in Section 2.3 and Section 2.4, numerous others were applied to enhance the trustworthiness of the research process and ultimate findings. Regarding data collection—and given the importance of rapport between interviewer and interviewee [61]—the quality of data was supported by the pre-existing relationships between the first author and all participants [62]. In addition, all interviews were undertaken in a private, quiet location at the training ground of the player’s club to aid comfort and openness. Of course, these advantages had to be balanced with measures to protect against any imbalances of power and the limitations of familiarity (e.g., the provision of socially desirable responses). Specifically, such issues were mitigated through the retrospective nature of the interview (i.e., the first author had no live management or selection influence as all participants were no longer academy players) as well as adherence to the components of ethical research by Hewitt [63]. The data collection process was further supported by a pilot study with two athletes who met the same inclusion criteria as detailed in Section 2.2 (M = 23.4). This work led to adjustments to the interview guide, with specific prompts altered and some jargon removed from questions.

Regarding data analysis, trustworthiness was enhanced by the first author’s use of a reflexive journal to document methodological and analytical considerations, the rationale behind decisions, and the interaction of the research team’s assumptions and biases [64]. In addition, the second, third, and fourth authors acted as critical friends across the full analysis. In particular, the fourth author provided critical feedback on selected procedures, while the second and third authors focused primarily on the use and outcomes of these procedures. As an accepted approach at the time of data collection, member checking was also used by returning transcripts to participants for them to assess the extent to which these accurately, fairly, and respectfully reflected their experiences [61]; a process which resulted in no significant changes. While a request for further member reflections could have added an extra dimension to our analysis [65], the checking process provided a degree of assurance on data fidelity.

## 3. Results

Addressing the first aim of the study and considering the different experiences of challenge through the pathway for each participant, Figure 1 shows the graphic timelines that were drawn by participants prior to the collection of interview data. They show the overall trajectory of athletes, representing their lived experiences of development and performance.

Across the sample it appeared that, regardless of birthdate in the selection year, there did not appear to be a pattern in the volume or intensity of the challenges faced on the journey to the professional game. Importantly however, this did not appear to be the case for players that were subsequently released. It appears that players born in Q1 and subsequently released experienced a challenge-free journey prior to academy entry at 16 years of age. Players born in Q4, and subsequently released, plotted their experiences in a similar manner. This contrasted significantly with the experiences of retained athletes with first and fourth quartile birthdates, who plotted a consistent series of bumpy challenging experiences prior to and through the academy system.

### 3.1. Player Perceptions of Challenge

As athletes progressing through the talent system, all identified a variety of challenges such as selection dynamics, peer to peer competition and increased stress from managing competing demands. Many of these challenges were associated with maturation dynamics. For example, consider the experience of these retained players:

I was tiny between 14 and 16. When I turned up at the academy at 16, I was 70 kg and still very small…I was always very small all the way until I was 16 or 17, that was when I actually really grew… I would never be able to physically dominate anyone at all. The only hope I had was to use my feet and pace which I think really, really helped and it’s probably why I ended up at 9 I think (Player 1: Q1—Retained).

It was only the fact that at 18 or 19 I found a bit of pace that kind of gave me that X factor to try to compensate a little bit for not being the strongest or the most physical. Physicality is one thing that has always been brought out with me in any review (Player 2: Q1—Retained).

I wasn’t physically muscular I don’t think I was strong I think compared to the others but I was quite tall and slim but I wasn’t massive, I don’t think I stood out from the crowd in any manner I was just a bit taller or you know probably in the top third of height—things like that at that age (Player 7: Q4—Retained).

For all eight participants, awareness of maturational status was a significant feature of their journey. These perceptions were emotionally laden and appeared to influence a wide range of behaviours. Despite self-identified, later-maturing players being present across quartiles, those who were retained did not express a perception of disadvantage, irrespective of birth quartile. Indeed, whilst reflecting on the consequences of later maturation, players appeared to perceive this as an enabling factor and something to work with over the long term. This contrasted with many released athletes who experienced early advantages through maturation:

I was bigger so I could run through people and get around people. It was easier to play because I was a little bit bigger. Skillset-wise I seemed to be a little bit behind (Player 8: Q4—Released).

It seemed seamless to me [transition into senior environment] to be honest I think it was because I was brought in to play in the second team games and then you come to some of the first team training sessions as well and then eventually, they brought you in full time… it was a good transition, it was easy (Player 4: Q1—Released).

I was bigger and taller than a lot of them, that’s the main bit. I had always been taller than everyone my own age (Player 5: Q4—Released).

As players progressed through the academy, significant differences emerged in their ability to deal with challenges. Across the retained group, players appeared to have the ability to utilise and reflect on past experiences of being disadvantaged. This perspective-taking appeared to influence perceptions of competence and control:

I knew I was better than players I was playing with at school level, but then you come somewhere like (club)… you know you are far from where you think you are. I kept my head down and worked hard, but I never felt like I didn’t deserve to be in the academy. I sort of felt that I deserved a chance to be in it and give it a shot, but when you get here you sort of realise there are 18-year olds who are way more physical… but that is good. At 16 you strive because you think I have got to catch him up, you know, it gives you goals (Player 1: Q1—Retained).

There was like older guys there as well… so we had 18/19-year-olds who were a lot more physically developed and experienced and better players than us so we were exposed to that and trained with that day in day out, at times like it was difficult… I had to deal with some right XXXX and eventually you start to find your way (Player 6: Q4—Retained).

I was still very small, I was still told probably too small to be a rugby player… it was never a thought of mine to be a professional rugby player but I was always going to be a small skinny player as far as I was concerned (Player 7: Q4—Retained).

In contrast, the released group struggled to cope with the increased range and intensity of challenges they faced. The response to these challenges was often perceived outside the player’s locus of control, with problems attributed to external factors. This contrasted with the retained group who saw challenges as obstacles to overcome by deploying a range of psycho-behavioural resources. As a consequence, released players appeared less equipped to cope with and learn from challenging periods:

It was frustrating that I could not do things that I used to do at 14/15 (years old), running and scoring plenty of tries, but my game changed a lot and I turned into a very different player due to that and it was frustrating (Player 8: Q4—Released).

It was quite scary because I had not played much in senior rugby, I just did not really know… so it was just quite scary not knowing where I was going to be and not knowing what I was going to do… I just lost direction (Player 3: Q1—Released).

Obviously, it was a lot more physically demanding and nothing you were sort of used to before. It was really tough… Just a lot more intense, a lot more volume with the actual rugby skill development and the strength and conditioning development. I’d never had it before, wasn’t really expecting it either (Player 4: Q1—Released).

The first couple of months it really p*ssed me off. You feel like you are standing still and you are desperate to play at that age, and then I remember just speaking to [brother] and my old man, and he just said ‘work hard and make sure that when you do get a go, you are ready to go’… it was then a case of looking at it from a different angle and saying I need to keep working at my passing and my kicking, the gym, the speed (Player 1: Q1—Retained).

As players continued their journey, these features seemed to become even more prominent, with overcoming a range of challenges seemingly a key differentiator.

### 3.2. Mechanisms Impacting Player Experience

Finally, in addition to exploring the individual players’ responses to various challenges, we also sought to understand the mechanisms that seemed to influence their overall experience.

#### 3.2.1. Nature of Commitment to the Sport

There appeared to be significant differences between the nature of the commitment to the sport between those who were retained and those released. Retained athletes seemed to engage and play with a focus on progression and enjoyment of the developmental process, rather than winning or domination of the game at earlier stages. In contrast, the released group seemed to heavily invest, from an early stage, with a focus on playing and winning matches, rather than engagement in other sports or training:

I moved there because it was the best team, the team I was with wasn’t that great and the mini set up was like fading out rather than like getting stronger and at the time (community club) had a strong mini section, so I joined that (Player 5: Q4—Released).

Released players also tended to focus solely on the outcome of selection in the short term, either for international rugby, or inclusion in an academy programme. This contrasted with retained players who seemed to focus on improving themselves rather than on an end goal of selection:

I was very rugby-focused not thinking too much about school. The next level for me was to get into the [club] academy and play for England at under 16s level, that was my driving goal (Player 8: Q4—Released).

The only thing I fixed on was decision making, two on ones, three on ones and obviously the backs were doing something different, all those skills I think that really benefited me and I remember thinking just focus on getting this stuff done... you’ll be better (Player 1: Q1—Retained).

#### 3.2.2. Nature and Influence of Support

Many of the retained group reflected on the use of experiences as a platform for reflection. Significantly, this seemed to be promoted by various supportive influences. This guiding of reflection seemed to be a key factor by which players were able to maintain perceptions of control during challenging experiences. In essence, this support seemed to be more facilitative, when compared to direct and ‘driving’ input of those who did not progress:

Mum and Dad used to make me clean my boots and that, I had one pair of boots and they had to last me for a season, so I was always told to polish them and look after them and make sure they did not split (Player 1: Q1—Retained).

I think by the time it came around to me playing they were supportive but not until I was about 16… I pretty much had to make sure that I got myself sorted for everything (Player 6: Q4—Retained).

Some subtle differences were observed between the retained and released groups in terms of the nature of the support from parents as they began to progress towards the transition to the senior game. In contrast to the retained players, the released group experienced far higher levels of support than those who were retained, indeed, something that did not appear to change, even as athletes progressed.

My Mum and Dad were so supportive that I didn’t need anything. They sort of volunteered and bought me wherever I needed to go—it was literally all for me (Player 8: Q4—Released).

#### 3.2.3. How Players Learned from Challenge

In addition, there appeared significant differences in the response of players to challenge and also how they learned from their experiences. Amongst the retained group, there appeared to be a greater perception of control during periods of challenge. As a result, players seemed to have the confidence to deploy previously developed skills and capitalize on the emotional experience of challenge. When this was not the case, especially amongst the released group, it appeared to be a barrier to long term progression. For player 8 reflecting on his early transition to the senior team and the changing perceptions of his earlier size advantage, led to the regret that he was unable to deploy the necessary skills to navigate the challenge:

You have got guys who were probably 20 kg heavier than me…I think a lot of it may have come down to confidence and I didn’t integrate well going into a first team environment… holding back a little bit more than I should have (Player 8: Q4—Released).

The differential response appeared to be a result of a lack of previous experience, reflection on, or development of the skills to cope with or learn from challenge. In contrast, amongst the retained group, players seemed to actively seek out challenging experiences. For example, player 1 deliberately chose to play in an age group beyond his chronological age as a means of increasing his challenge: “I was too young for that age group so at Sunday rugby I always played a year above” (Player 1: Q1—Retained).

We can also consider player 1′s perceptions of challenge as he progressed into the senior squad:

There was the likes of XXX and, a lot of the senior players who either were playing or had just retired and were coaching, really kind of nurtured me along the way… it was pretty tough period and I just kept focusing on getter better… yeah tough (Player 1: Q1—Retained).

## 4. Discussion

The specific aims of this study were firstly to generate a deeper understanding of the lived experience of the RAE, secondly to compare the lived experiences of early and late birth players, and finally to understand the mechanisms that influenced individual experience. We responded to criticisms of the existing body of research in RAE which has focused at the population level with limited use of qualitative methodologies to understand underpinning mechanisms.

### 4.1. Challenge

Whilst our exploratory approach set out to understand the impact of challenge in relation to RAE, what emerged was the impact of challenge irrespective of RAE. That is, later born players did not necessarily experience higher levels of challenge, nor did increased challenge necessarily lead to greater psychological growth [16,45]. As such, at the individual level, whilst players in this sample were drawn from the full spectrum of an age band, their relative advantage or disadvantage prior to the senior level seemed to have long lasting and significant effects. It appears that RAE is not in itself a mechanism. Instead, perceptions of and response to challenge seem more impactful than when an individual is born. Further, it suggests that, at the population level, whilst an early birthdate in the selection year is associated with early advantage, the degree to which this advantage persists is dependent on the ability of the individual to navigate/exploit future challenges. Indeed, these data suggest that the experience of significant challenge was an omnipresent feature of these athletes’ pathways, in the latter stages of an academy journey and whilst transitioning to the senior team, irrespective of birthdate [37].

### 4.2. Push and Pull Factors

Consequently, whilst relative age did not appear to be a mechanism in itself, there appeared to be three core factors that influenced player’s perceptions of control, confidence and overall perspective [38]. Participants who were better able to cope with and learn from the inevitable highs and lows of development seemed better able to orient their focus in a manner that would help them continue to progress. The ability to do this seemed to depend on skills that were developed prior to significant challenges and previous navigation of challenge often highlighted through maturational differences [66]. Moreover, the early development of skills impacted the player’s ability to cope with and learn from highly challenging experiences later in the pathway [67]. Data further highlighted the impact of these experiences and the skills deployed in the retained group’s reaction to challenge in comparison to the released group. This was consistently highlighted by the way each player was able to make sense of and process challenges as they occurred. This appeared to have significant impact on each athlete as they faced a series of emotionally laden challenges. Furthermore, this manifested in differences in the nature of the player’s commitment which continued as each one of them progressed [33]. Early advantages (often as a result of advanced maturation) seemed to drive an external focus (selection and winning). In contrast, early disadvantage seemed to promote a more internal focus on personal development. In turn, this suggests a reframing of RAE as a population-level effect, one that indicates a deeper phenomenon rather than having a direct effect.

By exploring the relative advantages and disadvantages of players at stages of their TD journey, against their later career success, we show that birth quartile number means very little without a deeper understanding of individual biopsychosocial context. What did appear critical for players to make the most of high challenge and subsequent emotional disturbance was the use of an appropriate range of psycho-behavioural skills [43,67]. Relative early advantage (experienced proportionately more frequently in earlier born groups) generates push-like effects. Push factors (pushing the player forwards), whilst allowing for early high performance relative to peers (and perhaps encouraging selection), seemed to retard later progress when missing skills were exposed [33]. Importantly, these push factors, whether they were high levels of parental input, or low levels of early challenge, were experienced across birth quartiles and seemed to have long lasting effects. In all but one case, the inability to overcome early push factors acted to prevent initial entry to the professional game and, even in light of eventual career status, only one player was able recover from deselection to play in the first tier of senior rugby. In contrast, those players who experienced more pull factors earlier in their TD journey (e.g., size disadvantages pulling them back), seemed to have more developmentally appropriate experiences that helped to prepare them for later challenges [42,44].

Of course, no research is without limitations. In this particular case, a common criticism of a pragmatic approach is that it risks provincialism, that is knowledge that is simply located in a particular context [68]. Indeed, whilst it is clearly not the prerogative of the epistemological approach used, or that of qualitative research in general, the same could be said for this study in adopting a relatively small sample of participants in a particular context. As a result, we ask the reader to focus on the principles underpinning our data and the possible transferability of findings to their own unique context [69]. Additionally, there is a risk that the relationship between participants and the first author as a leading professional in a rugby union academy programmes may have been a factor. This was mitigated by the individual participant timeline and interviews being conducted when participants had gone through the process of transition and retention and release status was established. There does however remain the risk of a player offering answers perceived as socially desirable. These risks were mitigated by adhering to the components of ethical research suggested by Hewitt [63] in acknowledging bias, developing rigor, a genuine level of rapport with participants, respect for their autonomy and the complete avoidance of exploitation.

## 5. Applied Implications

The evidence presented here raises the intriguing question of the extent to which various push and pull factors can be deliberately implemented in the experience of the athlete and at what point might they be appropriate. We would suggest that a sustained consideration (and balance) of push and pull factors should be a key feature of TD and, perhaps, participation environments [70]. This is especially the case for those aiming to support the development of athletes over the long term, not only developing junior career success (e.g., [71]). To be clear, this is not to suggest that an abundance of pull factors will always be a positive for overall development. Indeed, the large number of studies that explore the RAE show that greater numbers of pull factors may prevent athletes across sports getting selected in the first place [72]. In practice, these perspectives begin to challenge the hypothesis that relatively younger athletes will always benefit from playing against relatively older counterparts throughout development [8,20]. Consequently, for both research and practice we suggest that a rethink of our perceptions of RAE and its use as a metric in understanding TD is warranted. Quantitative methodologies have both offered insight into the impact of early advantages in terms of selection and the statistical consequences of challenge dynamics [1,8]. Across the domain, however, there is a need to complete more mechanistically focused research [73,74]. These insights are notable, not only considering the player’s initial retained or released status, but also the extent of their overall achievement. In taking a novel approach, we were able to consider cross-sectional data with the additional benefit of understanding a player’s long term career status. In the present sample, this allowed for demonstration that a number of these players progressed their careers to becoming the most elite players in the world. We would suggest that future research may benefit from adopting a similar approach, where cross-sectional data might be analysed considering long-term career status. In addition, we would suggest that research in RAE begins to move beyond further identification of RAE and advantage reversal in even more populations. Instead, a more granular consideration of the mechanisms at play is essential to truly understand the effect, an approach that has been taken in other areas of TD research [75]. From an applied perspective, this is essential if research is to make a difference in the real world and help the field think beyond simplistic solutions.

For many TD systems, it has been assumed that an appropriate target for selection has been a balance across quartiles, ensuring an appropriate number of Q4s are given opportunities [18]. We, nor any other researcher, can suggest what an ‘optimal’ balance of selection would look like, however, especially if a talent system was looking for an outcome marker of effective processes [73]. The evidence presented here should challenge simplistic narratives and the drive to ‘do something about RAE’. Instead, we suggest the need to focus on the individual in TD practice [76]. In addition, our data highlight the need for TD practitioners to have a central focus on the perceptions and needs of the individual athlete’s curriculum [77]. Our data also suggest that, whilst attempts to dampen, control and do-away-with the RAE are well intentioned, the unintended consequences of not exploring the complexity of this phenomenon may divert us from optimising TD practice. This is especially the case with top down systemic interventions that, by nature, require the simplification of complex processes [78]. Previous recommendations seeking to implement blanket strategies to mediate against disproportionately high pull factors seem overly simplistic. Strategies such as bio-banding or birthday banding may be easily implemented at the policy level but lack a holistic consideration of the biopsychosocial factors that influence relative advantage or disadvantage. As an example, there is growing recognition of social circumstances that may act as pull factors [79]. In essence, we are suggesting that if we are to offer truly practical implications to support the growth of a research-informed profession [80], the field should begin considering relative advantage or disadvantage on a holistic biopsychosocial basis, rather than using discreet indicators (e.g., maturation and/or relative age) alone. Notably, recent evidence has taken steps towards alternative approaches to levelling of the playing field with players being banded by technical competence [28]. In addition, there have been suggestions that coaches use a variety of methods for the grouping of players to provide a broad range of experiences for the player [28]. For practical purposes, we would suggest that coaches are better off implementing an approach built on individual periodising of challenge [36,81]. This could be achieved through greater flexibility of age bandings, allowing athletes to be offered appropriate competitive and training opportunities based on individual needs. As an example, it appears that a common practice for selection has become selecting players purely based on their birth quartile, with the assumption that Q3 and Q4 athletes will automatically possess a better psychological skillset. At a minimum, this paper should serve to challenge such simplistic narratives. Indeed, we would suggest that there is a core need for practitioners to begin focusing at a more individual level. The dynamics presented in this small sample, when compared to previous data [8,20], suggest the need for a far more individual approach in practice. This means that rather than resorting to blanket strategies, we need a more fine-grained approach to the grouping of individuals and management of challenge than previously advocated [25,27].

## 6. Conclusions

This study has considered the lived experience of relative age amongst a cohort of elite and near-elite rugby union players, analysed in light of eventual career status. Data presented clearly challenge beliefs held by the field of both researchers and practitioners. We suggest the need for a rethink of assumptions in the field, including the idea that RAE should be tackled with blanket policies. It is likely that RAE is a statistical outcome of challenge dynamics at the population level. We would therefore suggest a broader consideration of the dynamics of challenge, with a focus on the various push and pull factors that an athlete may be exposed to.

## Figures and Tables

**Figure 1 sports-10-00082-f001:**
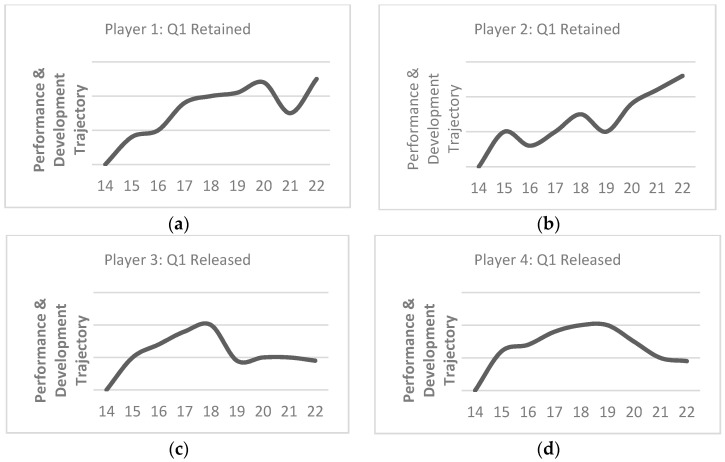
Graphic timelines of participants ((**a**–**h**), retained and released).

**Table 1 sports-10-00082-t001:** Participant information.

	Birth Quartile	Initial Career Status	Mid Career Status	Overall Career Status	Eventual Career Status
Player 1	Q1	Retained	Senior Test	Senior Test	Super Champion
Player 2	Q1	Retained	Senior Test	Senior Test	Super Champion
Player 3	Q1	Released	Championship	Championship	Almost
Player 4	Q1	Released	Championship	Championship	Almost
Player 5	Q4	Released	Championship	Premiership	Champion
Player 6	Q4	Retained	Premiership	Senior Test	Super Champion
Player 7	Q4	Retained	Premiership	Senior Test	Super Champion
Player 8	Q4	Released	Championship	Championship	Almost

## Data Availability

The data are not publicly available owing for the need to maintain the anonymity of participants.

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
