# Peer review of "Happy Birthday? Relative Age Benefits and Decrements on the Rocky Road"

_sports, 2022, doi:10.3390/sports10060082_

Round 1
Reviewer 1 Report
Dear authors,
I would like to congratulate you on your work. I have carefully analysed the manuscript and I suggest you to consider the following recommendations:
L47-48: description of the social construct should be implemented within the introduction
L80-82: Corrective adjustment procedures should be discussed around here (https://www.sciencedirect.com/science/article/pii/S1440244022001086#bb0125 ; https://www.tandfonline.com/doi/abs/10.1080/02640414.2021.1947618)
L170: SD = N.N. Please explain what N.N represents.
Author Response
We thank both reviewers for their timely and useful comments. Please see each addressed below:
Reviewer 1 comments
Thank you for your comments
L47-48: description of the social construct should be implemented within the introduction
41-44 - The inevitable chronological grouping of children as they enter the education system has been shown to promote early advantage through significant maturational differences for those born just before or after the academic cut-off date (11). This mechanism for selecting children continues as they enter organised sport and TS.
L80-82: Corrective adjustment procedures should be discussed around here
80-82 Thank you, this is a very useful point, Corrective Adjustment Procedures are another useful exemplification of the weaknesses of existing literature as applied to practice. In essence, better suited to be included as a population level strategy for creating fair and equal levelling of age bandings, accordingly we have included a reference to CAP in 84-90 and a short addition to the paragraph.
“It therefore seems that the current literature base has a number of key limitations. Those who are born earlier in the selection year are more likely to be selected for a TD system. It also appears that, at the population level, their relatively younger peers are more likely to continue through the TD system to elite performance. Yet, to this point, much of the extant literature has focused on ‘solving’ a variety of early advantage effects by focusing on levelling the playing field, for example: bio-banding [25]; age order shirt banding [26]; birthday banding [27] and corrective adjustment procedures (29).”
L170: SD = N.N. Please explain what N.N represents
179 - number of athletes included
Reviewer 2 Report
The main purpose of the work was to investigate the lived experience of relative age in talent development systems, compare the experience of early and late born players, and explore mechanisms influencing individual experiences.
The paper is generally well written based on sound literature, the results well presented and discussed with respect to the literature.
The work is written following the steps of the scientific method.
The study is well designed. However I have some minor comments I’d like to express.
At the beginning, try to define the inclusion and exclusion criteria for the analysis.
Provide a brief summary of the limitations of the evidence included in the review.
Perhaps the abbreviations should already be explained in the abstract.
It is worth adding a practical (appilcative) conclusion.
In my opinion, the conclusions should be more specific. not generalized, but this is only a suggestion.
Author Response
Thank you for you input, please find our responses below to your feedback
At the beginning, define the inclusion and exclusion criteria for the analysis
Sorry, we are unsure of the requirement here? The inclusion and exclusion criteria is provided in detail in the participants section. We are unsure what is meant by inclusion/exclusion criteria for analysis. Please advise.
Provide a brief summary of the limitations of the evidence in the review
If we understand this correctly, we have emphasised the limitations of current evidence base and the need for the approach adopted in this paper :
L80-83 “It therefore seems that the current literature base has a number of key limitations. Those who are born earlier in the selection year are more likely to be selected for a TD system. It also appears that, at the population level, their relatively younger peers are more likely to continue through the TD system to elite performance. Yet, to..”
L88-90 “As a result, we know that the RAE exists and that there is likely to be a reversal of advantage, but we don’t know why this happens. This is a key barrier for the practitioner seeking to optimise TD processes.”
Abbreviations should already be explained in the abstract
Thank you, L17 changed to: “Talent Development (TD)”
Is it worth adding a practical (applicative) conclusion
In my opinion, the conclusions should be more specific, not generalised, but is only a suggestion
Thank you, please see the addition of a conclusion. We hope this is appropriate, we would of course be very cautious offering a highly procedural set of recommendations, indeed, this is the type of approach that this paper would caution
L552: This study has considered the lived experience of relative age amongst a cohort of elite and near-elite rugby union players, analysed in light of eventual career status. Data presented clearly challenges beliefs held by the field of both researchers and practitioners. We suggest the need for a rethink of assumptions in the field, including the idea that RAE should tackled with blanket policies. It is likely that RAE is a statistical outcome of challenge dynamics at the population level. We would therefore suggest a broader consideration of the dynamics of challenge, with a focus on the various push and pull factors that an athlete may be exposed to.